# Global Perspectives on the Hepatitis B Vaccination: Challenges, Achievements, and the Road to Elimination by 2030

**DOI:** 10.3390/vaccines12030288

**Published:** 2024-03-09

**Authors:** Said A. Al-Busafi, Ahmed Alwassief

**Affiliations:** 1Division of Gastroenterology and Hepatology, Department of Medicine, College of Medicine and Health Sciences, Sultan Qaboos University, Muscat 123, Oman; 2Division of Gastroenterology and Hepatology, Department of Medicine, Sultan Qaboos University Hospital, Muscat 123, Oman

**Keywords:** hepatitis B, liver disease, WHO, vaccination, elimination, barriers

## Abstract

Annually, more than 1.5 million preventable new hepatitis B (HBV) infections continue to occur, with an estimated global burden of 296 million individuals living with chronic hepatitis B infection. This substantial health challenge results in over 820,000 annual deaths being attributed to complications such as liver cirrhosis and hepatocellular carcinoma (HCC). The HBV vaccination remains the cornerstone of public health policy to prevent chronic hepatitis B and its related complications. It serves as a crucial element in the global effort to eliminate HBV, as established by the World Health Organization (WHO), with an ambitious 90% vaccination target by 2030. However, reports on global birth dose coverage reveal substantial variability, with an overall coverage rate of only 46%. This comprehensive review thoroughly examines global trends in HBV vaccination coverage, investigating the profound impact of vaccination on HBV prevalence and its consequences across diverse populations, including both high-risk and general demographics. Additionally, the review addresses the essential formidable challenges and facilitating factors for achieving WHO’s HBV vaccination coverage objectives and elimination strategies in the coming decade and beyond.

## 1. Introduction

### 1.1. Overview of Hepatitis B and Its Global Impact

Decades of progress in the fight against hepatitis B (HBV) have produced effective screening tools, preventive vaccines, and antiviral medications, significantly enhancing patient outcomes [1]. Despite these advancements, HBV remains a critical global public health threat, causing both acute and chronic liver diseases and contributing substantially to the worldwide disease burden and mortality [2]. According to the Centers for Disease Control and Prevention, HBV Complications rank as the seventh-highest cause of global mortality [2]. 

Without intervention, individuals with chronic hepatitis B (CHB) face a lifetime risk of developing cirrhosis, liver failure, or hepatocellular carcinoma (HCC), ranging from 15–40% [3,4,5]. Correspondingly, HBV attributes to an estimated 29% of cirrhosis-related deaths worldwide [6]. Recent statistics indicate that approximately 50–80% of HCC cases globally result from HBV infection, making it the sixth most common cancer worldwide and the fourth leading cause of cancer-related deaths [7]. Moreover, about 90% of infants born to mothers testing positive for hepatitis B surface antigens (HBsAg) or e antigens (HBeAg) will develop chronic infection, posing a significant global public health challenge [8,9].

In 2019, the World Health Organization (WHO) estimated an alarmingly high global CHB infection prevalence, affecting around 296 million individuals worldwide [10]. That same year witnessed roughly 1.5 million new HBV infections, with HBV being responsible for an estimated 820,000 deaths, primarily attributed to complications like cirrhosis and HCC [10]. A 2022 modeling study by the Polaris Observatory group estimated the global CHB prevalence at 3.2%, equivalent to 257.5 million cases (Figure 1) [11].

Based on the same study, it is estimated that 84 million individuals, constituting 32% of the infected population, qualify for treatment in 2022. Although diagnostic testing for HBV has been available since the early 1970s, only 36 million individuals, accounting for 14% of the infected population, are believed to have received a diagnosis. Among these 36 million, merely 6.9 million (19.5% of those diagnosed and eligible, 8.1% of the total population who are eligible for treatment) currently receive antiviral treatment (Figure 2) [11].

This disparity is evident, as the majority of diagnosed and treated individuals cluster in developed regions. In contrast, sub-Saharan Africa and specific parts of Asia bear an unequal burden, grappling with heightened rates of prevalence and challenges linked to restricted access to vaccinations, diagnostics, and treatments [11]. Furthermore, the prevalence of CHB and new infections tends to be higher in lower-income or underdeveloped countries [12,13].

In 2022, the prevalence of HBV infection among children aged ≤ 5 years was estimated at 0.7%, corresponding to 5.6 million children (Figure 3) [11]. Notably, Nigeria, Indonesia, India, China, Angola, and Niger collectively accounted for 56.5% of all infections among this age group.

Despite the global decline in CHB infection prevalence, particularly among children under five due to infant vaccination programs, there are concerning disparities. While the WHO Western Pacific region (WPR) has a higher number of CHB infection carriers when compared to the WHO Africa region, the most significant percentage of children under five with HBV resides in Africa, with an estimated prevalence of 2.5% [14,15]. This persistent prevalence among African children under 5 indicates missed opportunities to control mother-to-child transmissions (MTCTs) of HBV [11,15]. While 85% of infants received the three-dose HBV vaccination before reaching one year of age, only 46% received a timely birth dose, and 14% received hepatitis B immunoglobulin (HBIG) as part of the complete regimen. Additionally, only 3% of mothers with a high HBV viral load received antiviral treatment to reduce the risk of MTCTs [11].

HBV transmission occurs through various routes, primarily via perinatal transmission, unsafe medical procedures, and unprotected sexual contact [16]. The chronicity of the infection varies geographically, with a significant proportion of infected individuals progressing to CHB infection, a long-term condition carrying substantial morbidity and mortality risks [17]. Despite advancements in vaccination-based prevention, barriers persist in achieving universal coverage and ensuring access to diagnosis and treatment, especially in resource-limited settings [18]. A comprehensive understanding of HBV’s epidemiology, transmission dynamics, and impact on affected populations is crucial for formulating effective controls and elimination strategies.

Efforts toward HBV control and elimination encompass a multifaceted approach, integrating vaccination programs, enhanced diagnostics, and affordable therapies. The WHO emphasizes the importance of vaccination as a cornerstone for prevention, aiming to expand coverage in order to reduce new infections [19]. However, achieving elimination targets demands not only robust vaccination programs but also improved access to diagnostics and antiviral therapies coupled with strengthened health systems [20].

### 1.2. Significance of Achieving HBV Elimination by 2030

Eliminating HBV by 2030 represents a critical global health milestone, aiming to significantly reduce the burden of this viral infection and its complications. In 2014, the Global Health Assembly assigned the WHO the task of supporting Member States in developing comprehensive strategies for preventing, diagnosing, and treating viral hepatitis [14]. Subsequently, in May 2016, the WHO established ambitious yet achievable targets to eliminate viral hepatitis as a public health threat by 2030 (Table 1) [14,21]. This goal relies on the following two key factors: service coverage and impact leading to elimination. Service coverage focuses on prevention and treatment, encompassing the following four key preventive strategies: Infant vaccination: Administering the three-dose HBV vaccine to infants.Preventing mother-to-child transmission (MTCT): Using either the HBV birth dose vaccine or alternative approaches.Ensuring blood and injection safety: Implementing protocols to minimize transmission through unsafe medical practices.Harm reduction: Implementing strategies to reduce transmission among high-risk groups like intravenous drug users.

This global strategy aims to make 90% of individuals with HBV aware of their infection and 80% of eligible individuals to receive antiviral treatment by 2030 [21,22,23]. This would result in a 90% reduction in the incidence of new infections and a 65% reduction in mortality. Achieving this necessitates comprehensive global efforts in prevention, diagnosis, treatment, and vaccination programs [22,24]. Aligned with the Global Health Sector Strategy (GHSS), new guidelines for the processes and standards for validating the elimination of viral hepatitis as a public health problem were provided by the WHO in 2021. According to those guidelines, the WHO suggests using absolute impact targets to validate elimination at the national level instead of the relative reduction targets initially defined in the 2016 GHSS. The leading impact indicators and targets for measuring the elimination of HBV new infection are ≤0.1% HBsAg prevalence in those aged five years or under.

Reducing new HBV infections is paramount, necessitating multifaceted strategies such as expanding vaccination coverage, especially in high-risk regions, and implementing effective prevention programs which address various transmission routes [22]. Two seemingly achievable goals are implementing the three-dose infant vaccine regimen and preventing MTCTs [25]. While most countries have incorporated HBV vaccinations into their national immunization programs, scaling up harm reduction practices among high-risk populations remains a crucial yet challenging aspect of combating HBV transmission [22].

Reducing mortality rates from HBV-related complications significantly contributes to its elimination [26]. Deaths from chronic HBV infections, including liver cirrhosis and HCC, remain alarmingly high [17]. However, increased access to diagnostics, timely antiviral therapies, and an improved healthcare infrastructure can significantly mitigate these adverse outcomes [17,20]. It is estimated that the implementation of the WHO 2023 elimination strategy would prevent 7.1 million deaths between 2015 and 2030.

Efforts toward HBV elimination offer not only health benefits but also substantial economic advantages. Numerous studies suggest that successful elimination could save healthcare systems billions annually via the reduction of disease management and complication costs [19]. However, achieving the 2030 elimination goal requires a critical reassessment of strategies and a renewed global effort [21,27]. The potential need to adjust elimination targets for practicality in regions grappling with high burdens of chronic HBV infections could be crucial [11]. Therefore, achieving global elimination by 2030 necessitates a two-pronged approach as follows: the rigorous and continuous evaluation of progress toward existing goals and the concurrent development of new, practically aligned strategies [19].

Despite being an ambitious task, the global elimination of HBV by 2030 requires considerable work in the coming decade. The review article aims to evaluate the pivotal role of HBV vaccination as a primary tool in realizing the ambitious goal set by the WHO to eliminate HBV by 2030. It focuses on analyzing the impact, effectiveness, and challenges associated with widespread HBV vaccination programs. The article likely discusses the vaccine’s ability to reduce transmission rates, prevent new infections, and contribute significantly to the WHO’s objective of lowering HBV-related morbidity and mortality. Additionally, it may explore strategies to enhance vaccine coverage, address barriers to immunization, and propose comprehensive approaches to harness the vaccine’s potential in achieving the global HBV elimination targets set for 2030.

## 2. Methodology

The review utilized comprehensive search strategies across multiple databases (PubMed, Scopus, Web of Science, ScienceDirect, and Embase) in order to gather the relevant literature, using keywords such as “Hepatitis B infection”, “Hepatitis B vaccine”, “Hepatitis B elimination”, and “WHO 2030 targets”. It collected a total of 412 relevant papers. The gathered studies, articles, and reports focus on HBV vaccination, global elimination efforts, WHO’s 2030 targets, vaccination barriers, and strategies for eradication, totaling 268 papers. The exclusion criteria may have omitted papers not directly related to HBV vaccination, studies beyond the WHO’s 2030 targets scope, or those lacking relevance to global elimination strategies. The extracted and synthesized data encompassed vaccination efficacy, barriers to immunization, global progress toward elimination, and strategies for achieving the WHO targets. A critical analysis of the gathered data identified trends, challenges, successes, and gaps in HBV vaccination programs, highlighting their role in meeting the WHO’s 2030 targets for HBV elimination. Integrated findings from selected studies created a cohesive narrative, emphasizing the pivotal role of the HBV vaccine in achieving the global elimination goals by 2030.

## 3. Hepatitis B Vaccines: Evolution and Effectiveness

Combating HBV infection involves three main strategies: treating chronic infections, eliminating transmission routes, and immunizing vulnerable populations [28]. Vaccination, arguably the most effective method, forms the cornerstone of the WHO’s strategy to eliminate and ultimately eradicate HBV. Since the 1990s, HBV prevention and treatment through HBV vaccination have been the subject of extensive research work. This section delves into HBV vaccine development, safety, adoption, and economic impact over the past three decades.

### 3.1. From Plasma to Recombinant: A Paradigm Shift

The first commercially available vaccine, launched in 1982, relied on extracting antigenic elements from the plasma of asymptomatic chronic HBV carriers [29,30]. Extracting the antigenic hepatitis B surface antigen (HBsAg) from the plasma of asymptomatic carriers formed the manufacturing process. This process involved ultracentrifugation and chemical decontamination, including formalin treatment to eliminate blood-borne pathogens [31]. The recommended three-dose schedule over six months proved highly effective and safe, establishing the benchmark for subsequent vaccines [29].

However, there were limitations. First, a theoretical but unproven risk of blood-borne infections like HIV raised public concerns, impacting vaccine acceptance [32]. Second, relying on chronic carriers as a source of HBsAg became unsustainable due to the vaccine’s success in reducing infection rates. These factors spurred the development of recombinant HBV vaccines [29]. In 1986, DNA recombination technology in yeast allowed for the synthesis of an HBsAg-like protein, leading to the FDA-approved recombinant vaccine [33]. Similar in efficacy and safety to its predecessor, it too required a three-dose, six-month schedule. These vaccines could provide long-term protection for over 30 years [30].

The latest advancement arrived in 2018 with HEPLISAV-B^®^, specifically designed for adults ≥18 years. This “combo” vaccine incorporates recombinant HBsAg with an adjuvant that amplifies the immune response, enabling a potent two-dose regimen instead of the three doses required by earlier versions. Studies consistently demonstrate its superiority, achieving an impressive 96.6% response rate after just one month, compared to only 24.0% for the conventional vaccine [34].

### 3.2. The Efficacy and Safety of HBV Vaccination

The HBV vaccine’s effectiveness can be measured in two ways: its ability to induce protective antibodies (anti-HBs) and its success in preventing MTCTs. Recombinant and combo vaccines show efficacy and safety comparable to the conventional version [33,34,35,36]. Across diverse vaccination schedules, they consistently achieve seroprotection levels exceeding 95% in healthy individuals.

The global adoption of the HBV vaccination has yielded remarkable results, with the worldwide prevalence in children under five plummeting from 4.7% in the early 1990s to 1.3% by 2015 [29]. However, studies indicate a decline in seroprotection among older adults (>40 years), highlighting the need for further research and tailored approaches for this population [37].

All HBV vaccines have undergone rigorous safety assessments, with over a billion doses administered since 1982, demonstrating an excellent safety profile [31,38,39,40]. Adverse events are typically mild and transient, such as injection-site reactions. Severe reactions are exceedingly rare, and studies have debunked unsubstantiated claims linking HBV vaccination to conditions like multiple sclerosis [41,42]. The HBV vaccination is demonstrably safe for pregnant and lactating women, low-birthweight infants, and even HIV-positive individuals, further expanding its reach and impact. Notably, the only contraindication is a known hypersensitivity to any vaccine component or a history of severe reactions.

### 3.3. Concerns with HBV Vaccination

#### 3.3.1. Durability of the Immune Response

The duration of immunity following the HBV vaccination is not definitively determined. However, in individuals, including both adults and children, who complete the initial vaccination series, attaining an anti-HB level of 10 mIU/mL or higher within the first two to three months following the final dose indicates a strong immune response effective against both acute and chronic HBV infections [43]. Studies show that this primary protection lasts for decades in most immunocompetent individuals [44]. Even when anti-HBs levels dip below 10 mIU/mL, protection has been documented for up to 30 years [45]. Follow-up studies reveal a natural decline in the concentration of anti-HBs over time, with a steeper drop in the first year and a slower decline later [46,47]. This pattern holds true, regardless of the initial peak antibody level. Other factors like age, weight, sex, and smoking history at the time of vaccination can also influence the vaccine’s durability (measured by anti-HBs levels above 10 mIU/mL) in immunocompetent individuals [48,49]. However, the effectiveness of HBV vaccination is not only linked to the induction of anti-HB antibodies but also involves the activation of memory B and T cells [45,50]. Therefore, even if anti-HBs levels decrease, that does not necessarily mean one is unprotected. The immunological memory of the HBsAg antigen can persist even when antibodies are no longer detectable [51,52,53,54]. In light of current scientific evidence, various advisory groups do not advocate for routine booster doses of the HBV vaccine in fully vaccinated, healthy, and immunocompetent individuals. This is because the majority of previously fully vaccinated individuals with anti-HBs antibody concentrations of 10 mIU/mL or less exhibit an anamnestic response when exposed to HBV or given a booster dose, indicating sustained protection by memory B and T cells [51,54,55,56,57,58]. However, there are exceptions where a booster dose might be recommended, such as for immunocompromised individuals or healthcare workers who are at a higher risk of exposure [51,52,53,54,55,56,57,58,59,60].

#### 3.3.2. Vaccine Escape Mutants

HBV vaccines and HBIG have effectively prevented infection before and after exposure [61,62]. However, the emergence of vaccine escape mutants poses challenges to the efficacy of these preventive measures. Studies highlight that mutations in the S gene of HBV, particularly in the α-determinant, can be selected under the immune pressure exerted via vaccination, especially when HBIG is administered [62]. The most common escape mutant, G145R, involves a glycine-to-arginine change at the amino acid position 145 [61]. Despite concerns, it is crucial to note that the baseline prevalence of these mutants varies geographically. In Taiwan, for example, the prevalence of α-mutants was 7.8% in HBsAg carrier children, remaining around 20% during the initial 15 years of the universal mass vaccination program [63]. Contrarily, over the last decade, there has been no substantial increase in vaccine escape HBV mutants among Taiwanese carrier children, with no evidence of the extensive spread of the virus [64]. This lack of escalation is attributed to the weakened nature of the mutant virus [64]. Similar findings in Italy suggest that these mutants may not pose a significant threat to ongoing control efforts [65]. Furthermore, studies reveal that current HBV vaccines can protect against infection with these mutant viruses, supporting the continued use of existing vaccines [66]. While these findings shed light on the challenges posed by vaccine escape mutants, the overall effectiveness of the HBV vaccination remains significant in preventing HBV infection. In addition, current HBV vaccines demonstrate efficacy in protecting chimpanzees from infection with these mutant viruses, thus supporting the continued use of the existing vaccines [67,68].

#### 3.3.3. Impact of HBV Genotypic Variability

HBV exhibits significant genotypic variability, categorized into ten distinct genotypes (A-J), possessing unique genetic characteristics [69,70]. This diversity manifests in geographically specific distributions, with certain genotypes dominating particular regions. For instance, genotype D is prevalent in Asia, while genotype B holds sway in Europe and North America [70]. This heterogeneity presents a challenge to vaccine efficacy, as current vaccines target a specific region of the viral surface antigen (HBsAg)–the “α-determinant” [37]. The challenge lies in the potential for reduced vaccine effectiveness against non-targeted genotypes, therefore hindering the achievement of universal protection.

Furthermore, the varying prevalence of HBV genotypes across regions can impact the overall efficacy of vaccination programs. Studies suggest that vaccines based on the A2 genotype may not offer complete protection against infection with non-A2 genotypes [71,72]. This underscores the intricate interplay between genotypic diversity and the potential for occult HBV infections, where the virus persists despite vaccination efforts [72,73,74]. Despite these challenges, large-scale studies reassure us that the currently available HBV vaccines remain highly effective against all genotypes, demonstrating a 94–98% protective rate against CHB infection for at least 20 years [70,75]. Moreover, the successful implementation of vaccination programs has led to a substantial decrease in the HBV carrier rate and HBV-related morbidity [62].

#### 3.3.4. Vaccine Non-Responders

Individuals who develop an anti-HBs titer below 10 mIU/mL after complete vaccination are classified as “vaccine non-responders”. This blunted immune response, often attributed to impaired T cell activation or the recognition of the HBsAg antigen, presents a significant challenge [76]. Globally, an estimated 10% of vaccine recipients fall into this category, highlighting the need for effective identification and management strategies [77]. Key risk factors include older age at primary immunization, obesity, smoking, comorbidities (diabetes mellitus, chronic kidney disease, chronic liver disease), HIV infection, and immune suppression [76,77,78,79,80,81,82,83,84,85]. Identifying these risk groups is crucial for planning follow-up screenings for anti-HBs levels and administering booster doses if needed. 

In studies of individuals over 60, only 45.7% developed anti-HBs antibodies [86]. Participants over 59 years old showed a 60% antibody titer greater than 10 mIU/mL seven months after the vaccination [87]. 

For patients with chronic kidney disease, seroconversion rates after three doses were 87.5%, 66.6%, and 35.7% for mild, moderate, and severe cases, respectively. Rates significantly improved after a fourth dose [88]. Patients with low glomerular filtration rates, higher creatinine, diabetes, and old age are less likely to seroconvert [89]. 

Patients with chronic liver diseases, especially hepatitis C infection, demonstrated a diminished response, with only 55% achieving seroconversion [90]. Genotype 1 patients exhibited a poorer response. The immune response correlated inversely with advanced liver disease, as measured by the MELD score [91].

In HIV patients, the seroconversion rate ranges from 18% to 72%, depending on one’s immune status. Those not receiving highly active antiretroviral therapy (HAART) exhibit response rates between 30% and 50%, while individuals on HAART experience an increased response ranging from 60% to 70% [92,93,94]. This response correlates directly with the CD-4 count and inversely with the viral load. 

A blunted response to the HBV vaccination can be augmented by interventions like additional booster doses, an increased vaccine dose, utilizing the intradermal route, or using combo vaccines [80,95,96,97]. One intervention, doubling the standard vaccine dose in cirrhosis patients, showed marginal success [98]. The intradermal route’s success relies on Langerhans cells and macrophages’ efficient antigen presentation, showing response rates of 69–100% [99]. A total antigen dose of 70–80 μg divided over 7–8 sessions or 5 μg weekly proved more effective than the standard 40 μg intramuscular protocol [95]. 

Vaccine advancements, like third-generation recombinant vaccines and adjuvanted vaccines, aim to improve responses, especially for immunocompromised and elderly populations [37,96,100]. Emerging vaccines present potential simplified schedules as alternatives to the traditional 0, 1, and 6-month regimens [39]. 

### 3.4. Vaccine for Healthcare Workers

Healthcare workers (HCWs) constitute a crucial target group for HBV vaccination, not only due to their susceptibility to infection but also because they may serve as potential sources of infection during exposure-related procedures. Globally, there exists a significant disparity in HCW immunization against HBV. Only 18% of HCWs in Africa reported receiving the HBV vaccination, contrasting with 77% in Australia and New Zealand [101]. A recent report highlights that almost half of HCWs in India remain unvaccinated. Conversely, Egypt boasts a higher adherence to immune prophylaxis, with nearly 82% of HCWs receiving the vaccination [20]. This discrepancy might be partially explained by Egypt’s inclusion of the HBV vaccination among its compulsory immunizations since 1992. However, the long-term retention of the protective effect in HCWs needs to be evident. A study revealed that 30% of previously immunized HCWs lost the protective anti-HBs titer (<10 IU) [102]. In light of these findings, making the HBV vaccination and regular anti-HBs checks mandatory for HCWs is imperative.

### 3.5. Cost-Effectiveness of HBV Vaccination

The potential cost-effectiveness of universal HBV vaccinations for high-risk groups has emerged as a key strategy for HBV prevention. A recent U.S. study by Chahal et al. compared incremental cost-effectiveness ratios (ICERs) between vaccination and test-and-treat strategies in select high-risk populations, including foreign-born Asians and Africans, incarcerated individuals, refugees, intravenous drug users, and men who have sex with men (MSM). The study found that the vaccination approach was more cost-effective than the test-and-treat approach, with ICERs of USD 6000 per each quality-adjusted life year (QALY) gained for the vaccination approach when compared to USD 21,000 per QALY for test-and-treat. These findings suggest that vaccinating all individuals within these high-risk groups may represent a valuable and economically viable public health intervention [103].

The cost-effectiveness of the HBV vaccination varies not only according to risk groups but also based on the administered vaccine type. A recent study demonstrated that the three-antigen HBV vaccine yielded improved health outcomes, with higher quality-adjusted life years (QALYs) and lower costs in select populations. Notably, for adults aged 18–64 years, adults with diabetes, and adults with obesity, the three-antigen vaccine was superior to the standard vaccine. Additionally, the analysis found the three-antigen vaccine to be cost-effective in adults aged ≥65 years when compared to the single-antigen vaccine, falling within commonly accepted willingness-to-pay thresholds (USD 26,237/QALY gained vs. USD 50,000–100,000/QALY gained) [104].

Numerous studies have investigated the cost-effectiveness of universal infant HBV vaccination, particularly in regions with intermediate or high HBV endemicity. The consensus across these studies indicates that universal infant vaccination represents a cost-effective strategy in such settings. Cost-effectiveness has been reported in Iran [105], China [106], and Vietnam [107]. On the other hand, the evidence becomes murkier in areas of low endemicity. Existing reports present conflicting findings, with studies suggesting ineffectiveness in some settings, such as the UK [108]. Alternatively, switching from adulthood vaccination to birth dose vaccination proved more cost-effective in Ontario, Canada. Moreover, incorporating the HBV vaccine into a hexavalent vaccine would save approximately USD 428,000 per disability-adjusted life year [109]. This lack of consensus necessitates further investigation to tailor policy recommendations to specific low-endemicity contexts.

Comparative mathematical models offer a valuable tool for evaluating the public health impact of interventions compared to maintaining the status quo. A recent Chinese study employed this approach to assess the national HBV immunization protocol. The study contrasted the two following scenarios by 2030: a “without vaccination” model and a model “with vaccination” incorporating the National Vaccination Protocol. This analysis revealed a significant decrease in HBV prevalence, dropping from 8.13% to 3.95% with the implemented protocol. Moreover, the study highlighted a substantial net benefit associated with preventing the MTCT of HBV, estimated at USD 12,283.50 per person [110].

Real-world studies are increasingly validating and refining the predictions of mathematical models in the field of HBV vaccination. A recent outcome study provides compelling evidence, demonstrating a superior effective vaccine protection rate (eVPR) for the two-dose HepB-CpG regimen compared to the three-dose HepB-Alum regimen. This finding translates to a lower cost per protection (CPP) for HepB-CpG (USD 331.31 versus USD 377.09). Moreover, the cost advantage is even more pronounced in individuals with diabetes (USD 367.57 versus USD 517.37), highlighting potential economic benefits for targeted vaccination strategies [111].

### 3.6. Recent Advances in HBV Vaccination

Recent advancements in HBV vaccination are pivotal in addressing challenges associated with first- and second-generation vaccines. Challenges persist, including the emergence of vaccine-escape mutants and low immune responses in high-risk groups, like older adults and smokers. Innovations like machine learning and blockchain are being explored to enhance the vaccine supply chain, logistics, and the overall coverage.

#### 3.6.1. Improving Vaccine Immunogenicity

A novel three-antigen (3A) HBV vaccine, “PreHevbrio”, demonstrated superiority over the standard single antigen vaccine in a phase 3 Multicenter RCT. PreHevbrio achieved higher seroprotection rates (SPR) of 90.4% and 99.3% after the second and third doses, respectively, compared to 51.6% and 95% for the standard vaccine. Additionally, it stimulated a 3.5-fold higher anti-HBs concentration than the standard vaccine [97]. PreHevbrio received FDA approval in November 2021 in the US, followed by EU approval in April 2022, and Canadian approval in December 2022 [97]. Similarly, Bahrami et al. explored a novel fusion plasmid DNA vaccine, demonstrating significantly higher anti-HBsAg titers than the standard HBV vaccine [112].

#### 3.6.2. Improving Vaccine Coverage

The pursuit of single-administration vaccine (SAV) protocols stems from the waning adherence to second and third doses in existing multi-dose vaccination regimens. SAVs target sustained antigen release, overcoming the limitations of intermittent booster approaches. One promising strategy involves incorporating stabilizing excipients into the vaccine delivery system. Among these, researchers have extensively explored the potential of alginate-chitosan-poly(lactic-co-glycolic acid) (PLGA) nanoparticles, recognizing them as a well-studied and potentially advantageous delivery system [113]. While the formulation demonstrated an adjuvant effect by generating higher antibody levels, uncertainties persist regarding the long-term immunological consequences of this delivery platform in humans. Notably, concerns have been raised regarding the potential tolerance induction with this specific dosing approach [114].

#### 3.6.3. Use of Alternative Routes of Vaccination

Administering an HBV vaccination orally offers numerous theoretical benefits, such as increased convenience, potentially reduced side effects, and enhanced cost-effectiveness [112]. However, formidable challenges hinder its progress. Firstly, the substantial size of the HBsAg obstructs efficient absorption through traditional oral delivery systems. Secondly, the stomach’s acidic environment significantly threatens the vaccine’s integrity, potentially jeopardizing its immunogenicity [115].

To overcome these challenges, researchers have achieved success in stabilizing HBsAg by using nanostructured silica SBA-15 [116]. Animal studies have reported that SBA-15 induces an immune response equal to or better than the injectable forms.

Efforts to develop alternative routes for HBV vaccinations have also explored the pulmonary route. This innovative approach involves delivering dry vaccine powder formulations deep into the lungs via specialized inhaler systems, achieving successful antigen delivery and deposition within the target tissue in mouse models [117].

#### 3.6.4. Therapeutic HBV Vaccines

Chronic HBV infection commonly encounters T-cell failure [118]. The evolving approach to treating HBV depends on enhancing host cytotoxic T-cell activity which targets HBV-infected hepatocytes [119]. 

The ongoing quest is to discover the ideal antigen that can successfully recruit T-cells with low toxicity to healthy liver cells. In a recent clinical trial, the GS-4774 vaccine, a complex antigen composed of HBs, HBc, and a partial HBV X protein, stimulated the immune system, but failed to provoke sufficient virus-specific CD4+ T-cells or B-cells to reduce circulating HBs levels [120]. In another study, an AI-powered model predicted the immunogenicity of 30 possible therapeutic vaccine legends. The model identified four epitopes with suitable profiles, exhibiting both immunogenicity and safety [121]. The hype surrounding AI in scientific studies suggests its potential to fill many gaps in the field of HBV prevention.

#### 3.6.5. AI Applications

Initiating the HBV transmission control cycle begins with delivering timely vaccinations. However, it is equally crucial to identify individuals at risk of a suboptimal immune response to the HB vaccine. One research group explored implementing machine learning to detect individuals at risk of vaccine failure and deliver a precision booster dose exclusively for those at high risk [122].

Likewise, a recent study utilized machine learning algorithms to identify the pre-vaccination ratio between two myeloid dendritic cell subsets (NDRG1-expressing mDC2 and CDKN1C-expressing mDC4) as a potential predictor of the immune response to a single dose of conventional HBV vaccinations. Specifically, individuals with a higher proportion of NDRG2-mDC2 cells exhibited a protective immune response after only one dose. In contrast, those with a predominantly CDKN1C-mDC4 population required two or three doses to achieve comparable protection [123]. If further research validates this finding, it could significantly impact optimizing HBV vaccination strategies by minimizing unnecessary booster doses, thus potentially reducing overall costs.

### 3.7. The Effectiveness of the HBV Vaccine in Different Populations

The HBV vaccine demonstrates substantial efficacy across diverse populations, particularly in high-risk groups. In pre-exposure scenarios, the vaccine plays a pivotal role in imparting immunity against HBV infection, especially for high-risk individuals such as healthcare workers and those in close contact with infected persons. Timely vaccinations post-exposure significantly diminish the risk of HBV transmission, providing a crucial protective layer against infection following potential exposure. Due to challenges in engaging and accessing populations at risk of HBV infection [124], a more practical and cost-effective approach emerged—the universal vaccination of newborns [125,126,127]. Consequently, the WHO endorsed and implemented the integration of the HBV vaccine into the Expanded Program on Immunization (EPI) [128,129].

Mass or universal vaccination campaigns, especially in regions with high prevalence, exhibit profound impacts, markedly reducing chronic HBsAg carriage rates and consequently lowering the burden of acute and chronic HBV-related diseases. This widespread immunization effort significantly decreases the incidence of HCC, a severe consequence of CHB infection, by suppressing the virus’s prevalence in the population and mitigating potential long-term complications.

#### 3.7.1. Vaccination for High-Risk Populations

##### Pre-Exposure High-Risk Groups

The HBV vaccine proves highly effective in safeguarding various high-risk population segments against HBV. For instance, it significantly benefits healthcare providers, fostering protective antibodies and long-term immunity [130], especially in high-risk groups, such as renal hemodialysis unit staff [131,132].

##### Patients with End-Stage Kidney Diseases

Studies reveal that while HBV vaccination is recommended for end-stage kidney disease (ESKD) patients on hemodialysis, this population often exhibits a lower immune response, with only approximately 50–60% achieving adequacy [133,134]. As chronic kidney disease (CKD) progresses, vaccine efficacy tends to decline [134]. The lower immunogenicity of the HBV vaccine in this high-risk population is attributed to various factors, including malnutrition, impaired immune function due to CKD, uremia-related immune dysfunction, and altered immune responses [135]. Response diminishes in patients with serum creatinine levels ≥4 mg/dL compared to those with lower levels (86% vs. 37%) [136,137]. Diminished responsiveness is associated with a glomerular filtration rate below 10 mL/min, people over 60 years of age, and diabetes mellitus [90,137,138,139]. Improved antibody responses occur among patients undergoing efficient hemodialysis [7,139,140]. Notably, coexisting hepatitis C virus infections may reduce vaccine effectiveness in patients undergoing maintenance hemodialysis [139,141]. 

The standard vaccination regimen often involves multiple doses (typically four) of the HBV vaccine before initiating hemodialysis [142]. However, achieving a robust immune response remains a challenge despite this regimen. Researchers have explored different approaches to enhance the vaccine response in this population, such as using higher vaccine doses, adjuvants, or alternative vaccination strategies [143,144]. Further research is needed to determine the most effective approach [138,145]. 

##### Post-Renal Transplant Patients

The immune response to the HBV vaccine is reduced following the renal transplant, with studies indicating decreased seroconversion rates and only 17-36% achieving protective antibody levels [146]. The vaccine’s efficacy diminishes as CKD progresses, posing challenges to attaining adequate immunity before renal transplant [134]. Post-transplantation immunosuppressive medications can further compromise vaccine effectiveness, influencing the patient’s ability to mount an adequate immune response [147]. Various strategies, such as alternative dosing regimens and routes of administration (e.g., intradermal), show promise in improving response rates [148]. Administering the HBV vaccination as early as feasible, preferably pre-renal transplantation, is crucial for ESKD patients due to the age-related correlation with the immune response [149] and the significantly heightened immunogenicity when administered prior to the transplant [150].

##### Men Having Sex with Men (MSM)

Studies show that the HBV vaccination remains highly effective in preventing HBV infection among MSM despite potential variations in immune response [151,152,153,154]. Randomized trials demonstrate an effectiveness of 80–88% in providing protection among MSM [154,155]. However, vaccine coverage remains a concern, lacking comprehensive data [155]. A systematic review suggests a possible association between the HBV vaccination among MSM and socio-demographic characteristics, behavioral patterns, and social-cognitive factors [155].

##### HIV Patients

HBV vaccination efficacy is diminished in HIV patients, leading to a reduced immune response [156,157,158,159]. Higher CD4 counts of ≥500/mm^3^ and lower HIV viral loads of ≤1000 copies/mL correlate with a more effective humoral response to the HBV vaccination among HIV patients [160]. The immune response might improve among patients with acquired immune deficiency syndrome (AIDS) undergoing highly active antiretroviral therapy (HAART) [156,157,158,159,160,161]. However, efficacy declines in HIV patients of older age, particularly after 40 years, or factors like obesity, stress, and smoking [162]. Strategies such as using double doses or booster shots have been shown to improve the immune response, enhancing the effectiveness of the HBV vaccine in HIV patients [156]. The recombinant HBV vaccine shows safety, efficacy, and cost-effectiveness in HIV-infected individuals [163] and should be offered to all patients regardless of their CD4 counts or viral loads.

##### Patients with Chronic Liver Disease

The HBV vaccination is generally safe in patients with chronic liver disease, displaying immunogenicity rates, although these are reduced (35%) when compared to the general population (95%) [164,165,166]. Studies emphasize the importance of assessing individual responses, revealing varied seroconversion rates ranging from 16% to 79% in clinical practice [164]. The severity of liver disease significantly influences the response to HBV vaccination, with rates differing between Child A (75.9%) and Child B (24.1%), highlighting the need for tailored approaches based on the stage of the disease [164,167]. Older age, diabetes mellitus, and cirrhosis are associated with reduced responses to the HBV vaccine [167,168]. Additionally, the underlying etiology plays a role; for example, patients with hepatitis C and alcohol-related chronic liver disease exhibit a low response to the hepatitis B vaccine [168,169,170]. Patients with chronic liver disease face an increased risk of encountering severe complications due to acute viral hepatitis A or B. This highlights the critical significance of vaccination for both hepatitis A and B in this patient population [166].

##### Post-Liver Transplant Patients

In liver transplant recipients, studies assessing post-liver transplant vaccination effectiveness vary significantly due to heterogeneity in recipient and donor types (cadaveric or living), vaccine type, adjuvant usage, vaccination protocol, response definition, and simultaneous HBIG or Lamivudine use [171]. Several studies highlight the challenge of achieving adequate immunity against HBV with a response rate as low as less than 30% [171,172]. The vaccination is notably more effective in patients who have undergone the transplant process due to HBV-related acute liver failure than those with CHB infection-related indications [171]. Immunoprophylaxis strategies, including HBIG and potent antiviral therapy (e.g., Entecavir, tenofovir), are often employed post-transplantation in order to prevent HBV recurrence [173,174]. Combining HBIG with antiviral medication proves more effective in preventing recurrence than HBIG alone [174]. However, this regimen’s critical weakness lies in its high cost and the risk of escape mutations related to HBIG. Therefore, the HB vaccine is being explored as a lower-cost substitute at many transplant centers, with varying reported effects [175,176]. Despite challenges, post-liver transplant HBV vaccinations may offer added immunity and are considered an active immunoprophylaxis strategy [176]. Some patients who receive the HBV vaccine develop sufficiently high hepatitis B surface antibody (HBsAb) titers to stop lifelong HBIG administration [175]. However, the efficacy might vary based on the regimen [176,177]. Studies analyzing different immunization schedules before and after the transplantation stress the importance of optimized vaccination protocols in these patients [177]. Recent reports highlight a limitation in the effectiveness of repeated vaccine doses to sustain adequate anti-HBs levels, primarily due to the emergence of escape mutations. This phenomenon poses a risk of diminished efficacy and increases the potential for the recurrence of HBV infections if used alone as a prophylaxis strategy [175]. Consequently, the strategy of post-liver transplant vaccination is not accepted as a reliable approach for preventing HBV recurrence in patients with chronic infections, thus necessitating the continuation of antiviral prophylaxis [171,172,178].

##### Mother-to-Child Transmission (MTCT) Settings

MTCTs, also known as perinatal or vertical transmissions of HBV, represent one of the primary routes of HBV spread worldwide (>85%), particularly prevalent in regions like Southeast Asia and China with high prevalence rates. The WHO estimated that in 2015, 65 million women of a childbearing age were chronically infected and at risk of transmitting HBV to their offspring [179], and 90% of infants infected with HBV at birth are at risk of chronicity [180,181]. MTCTs correlate with an increased risk of subsequent chronic liver disease and HCC [182]. Preventing MTCTs is a crucial strategy to meet the WHO 2030 goals and eradicate HBV [183]. Fortunately, the safe and effective hepatitis B vaccine has revolutionized MTCT prevention, significantly reducing transmission rates. Numerous studies demonstrate the remarkable efficacy of the hepatitis B vaccine in preventing MTCT [183,184,185,186,187]. 

Early studies reported significantly reduced MTCT rates with the prompt vaccination of newborns, achieving an MTCT reduction of 70–90% in HBeAg-positive mothers [181,184,185,186,187,188]. These findings support using vaccines in regions where pregnant women are not subjected to screening for HBsAg and HBeAg [189]. Subsequent research highlighted the synergistic effect of combining birth dose vaccinations with HBIG given within the first 24 h after birth, thus bridging the gap between HBV exposure and active anti-HBs production induced by the hepatitis B vaccine. This strategy yields even lower MTCT rates, reducing them by more than 90% and even approaching 98% in some cases [184]. The timely administration of dual immune-prophylaxis (the HBV vaccine and HBG) within 24 h of delivery is crucial, as delays have been linked to heightened infection risks among infants born to HBsAg-positive mothers.

Research indicates that maternal HBeAg positivity significantly impacts MTCT risks. In HBeAg-negative mothers, HBV vaccination alone can achieve impressive protection, with some studies reporting 0% MTCTs [188]. However, adding HBIG to the vaccination schedule offers additional protection for HBeAg-positive mothers, further reducing MTCT rates [187,190]. Even with HBV vaccination, approximately 3% of newborns from mothers who are carriers of HBsAg but negative for HBeAg may persistently contract the infection [126], indicating that defining infectivity by HBeAg in HBsAg carriers is not perfect [191].

The HBV viral load emerges as a critical predictor in assessing MTCT risk, surpassing the significance of HBeAg status. Studies consistently demonstrate a significant correlation between a high maternal viral load and an increased risk of MTCT, even in HBeAg-positive cases [192,193]. Recent studies suggest that higher maternal HBsAg levels are associated with an increased risk of perinatal transmission [193,194,195]. HBIG and vaccine failures occur almost exclusively in HBeAg-positive women with high HBV DNA levels (>200,000 IU/mL) and/or HBsAg levels above 4–4.5 log10 IU/mL [192,193,194,195]. Recognizing the paramount importance of early intervention, prophylactic measures during the third trimester have been recommended [196]. However, the decision to initiate NA prophylaxis is contingent upon the HBV viral load and the HBsAg level, with a consensus emerging around the threshold of 200,000 IU/mL and above 4–4.5 log10 IU/mL, respectively [192,193,194,195,196,197].

#### 3.7.2. HBV Mass Vaccination

After introducing the HBV vaccine, challenges arose when directing efforts toward high-risk populations like sex workers, homosexuals, or intravenous drug abusers [126]. Conversely, deeming universal vaccination for HBV in all newborns more feasible and cost-effective became apparent [127]. The WHO champions comprehensive hepatitis B vaccine programs for newborns as a cornerstone in the global effort to prevent the MTCT of HBV [198]. The WHO initiated universal vaccination in 1991 to prevent HBV transmissions during the perinatal period and early childhood, as these stages carry a higher risk of developing chronic infections [199]. This initiative emphasizes universal vaccination strategies to ensure maximum coverage and effectiveness. As of 2022, the WHO reported the nationwide introduction of the HBV vaccination program for infants in 190 Member States, showcasing widespread adoption [200]. The primary strategy involves providing the hepatitis B vaccine to all newborns, regardless of maternal HBsAg or HBeAg status. Screening pregnant women for HBsAg, with the addition of HBIG to newborns born to HBsAg-positive mothers, aims to enhance the protection against MTCTs, irrespective of the mother’s HBeAg status. 

Numerous studies substantiate the effectiveness of these mass vaccination programs, demonstrating a significant impact on reducing HBV infection incidence, especially in regions with a high disease burden [201]. Moreover, the overall success of these programs is underscored by integration into national immunization schedules, leading to increased coverage and sustained protection against HBV. Taiwan’s response to the substantial burden of CHB infections is characterized by a pioneering national vaccination program initiated in 1984. This marked a pivotal moment in Taiwan’s public health landscape and a strategic intervention to mitigate the widespread prevalence of CHB infections. By successfully incorporating HBV vaccination into routine immunization schedules, the program achieved remarkable progress in lowering the prevalence of CHB infection, HCC, and the mortality of fulminant hepatitis in vaccinated birth cohorts [201,202]. 

Different long-term cohort studies conducted in Taiwan have demonstrated the enduring effectiveness of this vaccination initiative over decades, with a decrease of more than 90% in chronic liver disease and HCC mortality and more than 80% in HCC incidence [201,202,203,204]. Taiwan’s experience serves as a model for other nations, showcasing the impact of a well-structured and enduring national vaccination program in mitigating the burden of CHB infection. 

Another study from Iran further supports Taiwan’s vaccination strategy on the effectiveness of the national HBV vaccination program 25 years after its introduction [205]. The study included over 100,000 individuals born before and after the program’s introduction. The results showed that the vaccination program was highly effective in preventing HBV infection, with a vaccine effectiveness of over 95%. Additionally, the study found that the program was cost-effective, with the benefits far exceeding the costs, making it a major public health success story that significantly improved the health of the Iranian population. From 1992 to 2015, HBV control via vaccinations in the Southeast Asia region made significant progress, preventing approximately 16 million CHB infections and 2.6 million related deaths during the same period [206]. 

Moreover, a study from China revealed that the vaccination strategy implemented over 20 years (1992–2012) was cost-effective and led to a significant decline in the incidence of HBV infections among children and adolescents [26]. The incidence among children aged 1–4 years decreased from 9.7% in 1992 to 0.3% in 2012, while the incidence among adolescents aged 15–19 dropped from 11.9% in 1992 to 1.1% in 2012. Globally, four distinct strategies for universal HBV vaccination in newborns are outlined in Table 2 [66,207].

The most straightforward approach involves administering the vaccine to all newborns without considering maternal HBV status. This strategy eliminates the need for screening pregnant women and bypasses the use of HBIG in newborns, resulting in the lowest associated costs. However, it is crucial to note that the efficacy of this strategy may be compromised, as indicated by various studies [184,191,208]. Conversely, the most comprehensive yet cost-intensive strategy is universally administering the HBV vaccine to newborns. This approach involves screening pregnant women and administering HBIG to newborns if the mother tests positive for HBsAg, irrespective of the HBeAg status. The choice of vaccination strategy is at the discretion of each country, guided by certain considerations, such as epidemiological factors, disease burden, public health system readiness, and economic constraints. It is imperative to emphasize that any selected approach for implementing an HBV vaccination program plays a pivotal role in controlling the prevalence of HBV within a given country. The decision-making process should be meticulous, considering various factors to tailor the strategy to the specific needs of each nation.

##### The Effect on Chronic HBV Infection

HBV vaccination programs globally have caused a dramatic decrease in CHB carrier rates, especially among younger generations. For instance, the HBsAg positivity rate in Taiwan plummeted from 19.7% in 1986 to 0.6% in 2015 [75]. The impact of HBV vaccination on the worldwide prevalence and incidence of CHB infections has been substantial. The implementation of routine infant vaccinations in over 180 countries has notably reduced global HBV transmissions and diminished CHB prevalence [209]. By 2019, the coverage of three doses of the HBV vaccine reached 85% globally, marking a significant increase from approximately 30% in 2000, highlighting the successful implementation of vaccination programs [63].

The global effectiveness of the HBV vaccine is apparent in the substantial reduction in CHB prevalence and incidence, supported by comprehensive vaccination programs and documented success in diverse populations [26,110,126,191,201,202,203,205,207]. In regions with lower endemicity, the prevalence of the HBsAg carrier after the vaccination may effectively decrease to zero [66]. This phenomenon signifies a promising trajectory toward the elimination and ultimate eradication of HBV within the population [66]. Furthermore, the research emphasizes the highly effective nature of current vaccines, providing a protection rate of 94–98% against CHB infection for at least 20 years [66]. The decline in reported acute HBV infections by approximately 90% since the introduction of HBV vaccination recommendations further underscores the vaccine’s impact, reducing the rate from 9.6 cases per 100,000 [210]. Additionally, recent findings indicate that HBV vaccination positively influences the survival of patients with chronic liver disease, further emphasizing the vaccine’s broader health benefits [37,164,166]. 

##### The Effect on Diseases Related to Acute HBV Infection

Research consistently demonstrates the HBV vaccine’s effectiveness in reducing diseases related to acute hepatitis B infections [211,212,213,214,215]. Studies reveal a significant decline in the incidence of symptomatic acute hepatitis B and fulminant hepatitis in vaccinated individuals as opposed to those who have not received vaccination [211,213,214,215]. The vaccine effectively reduces the risk of severe liver-related complications linked to acute infection, such as liver failure or chronic hepatitis [212]. Additionally, it substantially decreases the likelihood of hospitalizations and mortality related to acute hepatitis B [211]. In Italy, where universal HBV vaccinations were initiated in 1991, there was a notable decrease in CHB infection and the incidence of acute hepatitis B and hepatitis D [213,214]. However, it was observed that interrupting HBV infections through household contacts of CHB carriers, injection drug use, and iatrogenic procedures was still necessary to eradicate residual HBV infections in the country. This evidence underscores the vaccine’s crucial role in preventing the severe health consequences of acute hepatitis B, highlighting its importance in public health initiatives aimed at reducing the burden of hepatitis-related illnesses.

##### The Effect on Diseases Related to Chronic HBV Infection

Extensive research consistently supports the effectiveness of the HBV vaccine in reducing HBV-associated membranous nephropathy (HBV-MN). Several studies conducted in different populations indicate that vaccination significantly decreases the incidence of HBV-MN, following nationwide HBV vaccinations [216,217,218]. Additionally, the vaccine substantially reduces the risk of CHB-related mortality by over 90% over 30 years [202]. 

##### The Effect on Hepatocellular Carcinoma (HCC)

HCC stands as the most prevalent primary liver cancer, representing a leading cause of global cancer-related deaths. CHB infection remains a significant risk factor for HCC, contributing to approximately 50-80% of cases worldwide [7]. In areas with intermediate to high HBV endemicity, HCC predominantly affects middle-aged or older individuals; however, occasional cases are observed in children, usually linked to CHB infections acquired through maternal transmission [182].

The development of a safe and effective HBV vaccine has provided a potent tool for preventing HBV infections and any associated complications, including HCC. Vaccination programs globally have resulted in a substantial reduction in CHB carrier rates, particularly among younger generations [66]. For example, in Taiwan, the HBsAg positivity rate dropped from 19.7% in 1986 to 0.6% in 2015 [219]. As fewer individuals contract chronic HBV infections, the long-term expectation is a decline in the prevalence of HCC cases. This trend is already evident in certain regions, with younger populations exhibiting significantly lower HCC rates when compared to their older, unvaccinated counterparts [66]. 

Numerous studies have illustrated the remarkable effectiveness of HBV vaccination in reducing the incidence and prevalence of HCC. In Taiwan, the introduction of universal neonatal HBV vaccinations in 1984 resulted in a significant decline in HCC incidence among younger age groups, achieving an 80% decrease after 30 years [220]. Similar positive trends were observed in Shanghai, with a 49.2% reduction in HCC incidence in males and 51.9% in females 30 years after the initiation of their vaccination program [220]. Globally, studies conducted across various regions and countries consistently report substantial reductions in HCC incidence among cohorts born after the implementation of vaccination programs [221,222,223,224].

## 4. The Global Impact of HBV Vaccination Programs: Successes and Challenges

The HBV vaccine is the keystone intervention in combating the worldwide burden of HBV infection. As of the close of 2022, the WHO reported that an impressive 190 nations (97%) integrated birth doses into their EPI schedules for comprehensive coverage [200]. Infant completion rates for the three-dose hepatitis B vaccine regimen surged remarkably from 1% in 1990 to 84% in 2022 [200]. Furthermore, 113 Member States implemented a nationwide single-dose HBV vaccine administration program for newborns within the initial 24 h of life. 

The HBV vaccine boasts a remarkable track record, demonstrating a near 95% reduction in CHB infections [201]. Global HBV prevalence significantly dropped from over 10% in the 1980s to around 2.9% in 2020, resulting in millions of lives saved and the prevention of chronic liver disease and HCC cases [221].

The vaccine’s impact transcends individual protection, safeguarding entire communities and future generations from the devastating consequences of HBV infections. This herd immunity effect is evident in countries with high-coverage infant vaccination programs, observing a noteworthy decline in transmission among vaccinated and unvaccinated groups [50].

Recent large studies analyzing the positive impact of universal HBV vaccination reveal a global decline in HBV prevalence, especially in children under five [11,225], considering prophylaxis programs, notably infant vaccination. Mainland China, with the highest overall HBV prevalence, ranking 16th in HBV infections among ≤5-year-old children, attributes its success to a timely birth dose coverage of 90% or greater [11]. 

This remarkable progress can be attributed to several key factors. Foremost, the WHO established global goals to attain HBV elimination by 2030, underscoring the importance of timely birth doses to prevent MTCTs and the early horizontal transmission of the virus, the most prevalent cause of HBV transmission worldwide [10,11,14,23,221]. Increased awareness among healthcare providers and communities about the benefits of timely birth doses has contributed to higher acceptance and implementation [226]. Moreover, the significant reduction in vaccine cost (below USD 1 per dose), crucial financial and logistical support provided by GAVI since 2001, and the substantial rise in pentavalent vaccine coverage in GAVI-supported countries all played a significant role [227]. Despite overall progress, regional disparities persist. While the Western Pacific (90%), Americas (89%), and Southeast Asia (87%) regions exceed the global average, Europe (81%), the Eastern Mediterranean (80%), and Africa (75%) lag behind [200]. 

The WPR, with the early adoption of birth-dose vaccinations, stands as a testament to the program’s efficacy. Notably, the region’s HBsAg positivity rate dropped from 8.3% to a mere 0.93% between 2002 and 2015 [37]. Furthermore, HBV vaccinations effectively protect healthcare workers from occupational infection and subsequent chronic diseases [49,59,101,102].

Early adopters like Taiwan, Bulgaria, Malaysia, The Gambia, Italy, Spain, and the United States exemplify success through universal HBV immunization. Taiwan, in particular, witnessed a dramatic decline in HBV transmission, disease burden, and HBsAg positivity since its mass neonatal vaccination program in 1984 [9,75,126,191,201,202,204]. The annual HCC incidence among children aged 6-14 significantly decreased, solidifying the vaccine’s potential as a successful cancer-preventative measure [219].

The global adoption of HBV vaccination and enhanced coverage have yielded remarkable results in reducing the HBV burden and preventing HCC. Ongoing efforts to achieve universal coverage, particularly in regions with lower implementation rates, offer substantial potential for further alleviating the public health impact of this critical viral infection. 

Despite undeniable successes, meeting the 2030 WHO targets for HBV vaccination necessitates addressing substantial challenges associated with vaccine utilization. Numerous barriers impacting the effective expansion of HBV vaccine uptake have been identified [228,229,230,231,232], presenting diverse characteristics across countries and affecting different target groups. These barriers vary based on healthcare systems, caregiver awareness, cultural beliefs, and accessibility to vaccination services. These barriers can be categorized into different groups, including System Issues”, “User Issues”, “Service Provider Issues” and “Socio-Culture Factors” [231]. Key barriers include limited availability and accessibility to health-facility-based immunization, a lack of caregiver awareness, inadequate communication by healthcare workers, negative relationships with beneficiaries, high vaccine costs in the private sector, and challenges related to the time and place of vaccination [231]. Emphasizing the contextual nature of these barriers is crucial, recognizing that different countries are at varying stages of implementing the HBV vaccination [231]. 

Understanding the intricate challenges is essential to achieving the widespread HBV vaccination which aligns with the WHO framework. A comprehensive strategy involves addressing issues of affordability, enhancing healthcare infrastructure, eliminating sociocultural obstacles, prioritizing HBV within public health frameworks, and fostering effective communication and collaboration. These measures constitute fundamental building blocks for a future where the impact of HBV is significantly reduced, realizing the full potential of this life-saving vaccine. 

## 5. Progress towards Achieving the WHO Targets for 2030 HBV Elimination: 2016–2023

### 5.1. The Impact of the HBV Vaccination

Since its introduction in 1982, the HBV vaccine has revolutionized the battle against this insidious pathogen. As the WHO sets ambitious targets for hepatitis B elimination by 2030, the vaccine emerges as a cornerstone intervention with a profound impact on reducing the disease burden [14]. From 2015 to 2020, global coverage of the third dose of the HBV vaccine increased from 82% to 85%, while the administration of the birth dose increased from 38% to 43% [233]. Infant vaccination programs, particularly in high-endemic regions like Taiwan, showcase a notable decline in HBsAg positivity from 9.8% in 1984 to 0.6% in 2004, primarily driven by neonatal immunization [126,191,202].

The influence of the HBV vaccine extends beyond preventing chronic infection. Reducing carrier rates subsequently lowers the incidence of HBV-related complications, including HCC, a major cause of mortality [234]. Mathematical modeling estimates reveal that global HBV vaccination programs averted 210 million new HBV infections and 1.2 million HCC deaths between 2000 and 2015, translating to millions of lives saved and significantly reducing the healthcare costs associated with HBV-related complications [179,235]. 

### 5.2. The Feasibility of HBV Elimination by 2030

As defined epidemiologically, HBV elimination involves reducing HBV incidence to zero in specified geographical areas due to deliberate efforts. Presently, HBV elimination aligns with the 2016 WHO targets for viral HBV infections, focusing on controlling viral hepatitis by reducing its incidence, morbidity, and mortality to locally acceptable levels rather than absolute elimination [14,236]. The global initiative for HBV elimination by 2030, as outlined by the WHO, involves three key goals. These objectives include the establishment of a world free from HBV transmission, ensuring individuals with viral hepatitis access to safe, quality, affordable, and effective care, and eliminating viral hepatitis as a significant threat to public health by 2030, recognizing that complete elimination may not be entirely achievable. Additionally, the aim is to significantly reduce the incidence of chronic viral hepatitis, along with its correlated morbidity and mortality rates [10,14].

The feasibility of global HBV elimination is rooted in the virus’s characteristics, dependable diagnostic tests, and cost-effective strategies. Measures include implementing universal HBV immunization, antiviral treatments for highly viremic mothers in the third trimester to prevent MTCTs, HBV screening in blood donors, adopting safe injection practices, implementing stringent infection-control programs to reduce HBV infections, and providing antiviral treatments for patients with HBV infection [236]. Despite the ambitious task of achieving global HBV elimination by 2030, substantial work is imperative in the coming years. 

The available evidence suggests that the ambitious goal of achieving global HBV elimination by 2030 faces significant challenges. Despite the existence of tools to reach these goals since 2015, they remain insufficient or absent in various countries and regions, including high-income nations. Globally, HBV diagnosis rates are alarmingly low, averaging at only 8% [237,238]. Despite advancements in preventive measures and prevalence goals for HBV in specific regions, the research underscores the imperative for all areas to markedly enhance diagnostic rates and accessibility to treatment in order to achieve global targets [17,239,240].

Unfortunately, given the staggering number of individuals with CHB infection (257 million) and its devastating global impact (almost 900,000 annual deaths) [179], coupled with the current efforts to combat the disease, achieving these targets in the next remaining years appears unlikely, particularly in the resource-limited settings. The WHO has advocated for multiple key interventions to achieve these targets, yet implementation remains lacking in most locations [179]. 

Achieving 90% coverage for HBV immunization, including the HBV birth-dose vaccine, faces considerable challenges, especially in regions like Africa, where the current estimated coverage for the HBV birth-dose vaccine is merely 11% or even lower in resource-constrained areas [241]. Similarly, in low- and middle-income countries (LMICs), the significant challenge of raising antiviral treatment availability to 80% by 2030 is emphasized, especially considering that currently, less than 5% of individuals infected with HBV or HCV undergo testing or are enrolled in care and treatment programs [179].

In many LMICs, obtaining affordable access to viral hepatitis testing and treatment proves nearly impossible, acting as a significant impediment to the expansion of screen-and-treat interventions [238]. 

High-income countries face their challenges, as millions of undiagnosed and infected individuals, often from vulnerable populations such as intravenous drug abusers, homeless individuals, and undocumented migrants, pose difficulties in reaching and enrolling them in care. In various regions, particularly in LMICs, where the population is aging and mostly untested, there is a likelihood of a higher mortality from viral hepatitis in 2030 compared to 2015. Additionally, to track advancements toward the WHO elimination goals, it is imperative not only for countries to develop national hepatitis plans but also to implement surveillance systems for evaluating the occurrence and impact of liver disease. Such surveillance systems are either absent or inadequately developed in most LMICs [179]. 

While recognizing commendable initiatives by the WHO in collaboration with different stakeholders to initiate the elimination of viral hepatitis, an urgent need arises for developing and implementing realistic strategies tailored to diverse environments and specific populations. Addressing this urgency requires increased engagement from civil society, health policymakers, and funders, given the current insufficient dedication to combat viral hepatitis. In contrast to diseases like HIV and malaria, viral hepatitis faces substantial underfunding for both research and elimination efforts.

The WHO urges nations and regions to invest in eradicating hepatitis by integrating costing, budgeting, and financing elimination services into their universal health coverage plans. To achieve this goal, it is crucial to implement not only effective treatment but also comprehensive policies addressing infection prevention, political commitment, financial structures, stakeholder engagement, and healthcare system integration. Combining prevention and treatment is a viable strategy against viral hepatitis, but it demands significant investments in strengthening healthcare systems and providing a complete range of hepatitis services. The expected outcome of this investment includes producing economic advantages directly, indirectly, and across various sectors by saving lives and alleviating the financial strain caused by the disease on individuals, their families, and the counties [242]. 

The projected expenditure for key interventions in LMICs between 2016 and 2021 amounts to USD 11.9 billion, primarily driven by the costs associated with testing and treating chronic viral hepatitis [22]. Economic evaluations conducted in diverse regions highlight the cost-effectiveness of population-based approaches to testing and treatment, emphasizing the need for an annual strategic investment of USD 6 billion. This investment aims to prevent 4.5 million premature deaths by 2030 and more than 26 million deaths beyond that milestone. However, if medicines remain inaccessible and patent-protected in 13 LMICs, the cost could escalate to USD 118 billion [243]. 

To align with the WHO’s target of eliminating viral hepatitis, there is a need for a substantial increase in global diagnosis coverage from 9–20% in 2015 to 90% by 2030. Additionally, treatment coverage should progress from 7–8% in 2015 to 80% by 2030 [237]. In practice, successful treatment coverage relies on countries enhancing their efforts in implementing national plans for effective diagnosis and subsequent linkage to care. However, there are notable gaps in the existing policies. In a 2017 survey of all 194 Member States, approximately 70% (135 countries/regions) formulated national plans for the WHO elimination goals. However, fewer than 50% secured funding, and even in funded cases, the allocated amounts fell short of covering the entire plan [233]. High-income countries face ‘diagnostic burnout’, treating easily accessible patients while leaving marginalized high-risk groups, such as the homeless, prisoners, intravenous drug abusers, and a significant portion of the general population, undiagnosed. Consequently, 80% of high-income countries are not progressing towards meeting the viral hepatitis elimination goals by 2030, with 67% lagging by at least two decades [244].

In 2019, the Lancet Gastroenterology and Hepatology Commission, focusing on accelerating the elimination of viral hepatitis, provided an overview of the status of 11 policy indicators related to viral hepatitis in 66 heavily burdened countries and territories across different global regions [238]. The commission highlighted substantial failures in policy implementation across all the studied countries, mainly due to a scarcity of readily available and adequate national epidemiological data as well as the lack of government-funded screening and treatment programs for viral hepatitis. Another study, utilizing the commission findings, found a diverse spectrum of policy responses across countries and territories, with some having all recommended HBV measures in effect or development. Meanwhile, other countries performed inadequately, often with only a single policy in place, such as mandatory screening for transfusion-transmissible infections, including viral hepatitis [245]. Furthermore, it is noteworthy that many countries face a significant challenge as they lack comprehensive estimates regarding the potential economic repercussions of viral hepatitis on their respective populations [245]. This information deficit creates a barrier to understanding the full scope of the economic burden imposed by viral hepatitis within these nations. The absence of detailed assessments hinders the ability to formulate targeted strategies and allocate resources effectively to mitigate the economic impact of this health issue.

### 5.3. The Effect of COVID-19 on the 2030 Elimination Plan

The COVID-19 pandemic has had profound effects on global initiatives to eliminate hepatitis B and C by 2030. A multinational European Association for the Study of the Liver (EASL) survey reveals a significant impact across diverse regions [246]. Numerous elimination programs have experienced slowdowns or complete halts due to the pandemic, resulting in a potential one-year delay in hepatitis diagnosis and treatment. This delay could lead to an alarming increase in liver cancers and deaths attributed to HCV, with projections estimating 44,800 additional liver cancers and 72,300 deaths globally by 2030 [247].

The WHO’s elimination target of viral hepatitis worldwide by 2030 was severely affected by the divergence of attention and resources towards COVID-19. The necessary focus on the pandemic has led to doubts surrounding achieving the previously commendable goal [247]. To combat this setback, a collective reassessment and revision of global and national goals and action plans are imperative within the hepatitis community.

The economic downturn associated with the pandemic has posed challenges for committing new funds and jeopardized previously allocated resources for viral hepatitis elimination [248]. The long-term impact of COVID-19 on global healthcare systems underscores the need for the strategic utilization of resources invested in mitigating the pandemic’s effects. These funds could contribute to the reinforcement of surveillance and healthcare systems, ultimately enhancing viral hepatitis services [249].

### 5.4. Key Advancements in Achieving the 2030 Elimination Targets

Significant progress has been made since 2015 towards achieving the WHO’s ambitious 2030 goals to eliminate viral hepatitis. The WHO has been a key player in addressing HBV globally, providing essential tools and strategies tailored to each member state. The organization assists Member States in formulating national strategies, aligning them with specific needs and available resources [212,213,214,215,216,217,218,219,220,221,222,223,224,225,226,227,228,229,230,231,232,233,234,235,236,237,238,239,240,241,242,243,244,245,246,247,248,249,250,251,252]. Furthermore, the WHO plays a pivotal role in issuing guidelines for preventing, caring for, and treating HBV, which are essential for effective management and control. The WHO CHOICE also performs quantitative analysis, contributing to cost-effectiveness assessments for hepatitis prevention and treatment strategies. Additionally, the WHO has implemented a global reporting system for viral hepatitis, bolstering surveillance and data collection efforts worldwide. The organization formulates comprehensive guidelines for strategic information, establishing a robust framework to address hepatitis on a global scale. A study was conducted by the WHO in 2017 to assess how member states were responding to global viral hepatitis [233]. The findings indicated that 62% of Member States had formulated national plans for viral hepatitis, with 27% in the draft stage. Among those with developed plans, 58% had allocated funding for implementation. Furthermore, the study emphasized that engaging civil society increases the likelihood of having national viral hepatitis plans, dedicated funding, and established laws or policies addressing stigma and discrimination.

Significant advancements have been made in the immunization coverage against HBV infection. By the end of 2018, the HBV infant vaccine had been introduced across 189 Member States, attaining a worldwide coverage rate of 84% for three vaccine doses [241].

Historically, the WHO WPR had the world’s highest CHB infection prevalence, with various countries in this region, including the Philippines, Vietnam, and China, having prevalence rates exceeding 5%. In 2017, the WPR achieved a significant reduction in CHB infection prevalence, reaching 0.93% in 5-year-old children, meeting the WHO target [253]. Looking ahead to 2023, GAVI’s enhanced support for HBV vaccine birth-dose coverage could potentially avert up to 1.2 million perinatal infection-related deaths and prevent up to 1.5 million HBV cases over a span of 15 years [254]. 

Despite this progress, estimates indicate insufficient HBV birth-dose vaccine coverage globally, particularly in Africa, where only 11% of newborns receive the recommended dose within 24 h [241]. Civil society bodies like the World Hepatitis Alliance continue to play a pivotal role in engaging donors and stakeholders, ensuring an evidence-based approach to the response.

Examining Taiwan as a case study reveals substantial advancements toward the 2030 elimination goals. These include preventing vertical HBV transmission through immunization and antiviral treatments for highly viremic mothers, ensuring widespread access to potent, safe, and affordable HBV treatments, and initiating changes in reimbursement policies for HBV therapy. Taiwan’s proactive approach, including universal HBV vaccination since 1986, significantly reduces HBsAg carriage rates and expands treatment indications. The country aims to reach the WHO goals by 2025, five years ahead of the 2030 deadline [202]. 

Developed over a decade ago, the latest generation of nucleoside analogues for HBV has demonstrated enduring safety and effectiveness in preventing chronic hepatitis B-related liver complications, such as cirrhosis and HCC [255]. The introduction of generic medications has not only decreased treatment costs but has also expanded treatment accessibility in numerous countries. Additionally, a crucial stride towards enhancing global vaccine coverage involves the creation of polyvalent vaccine formulations encompassing the HBV vaccine for infants and children.

Advancements in HBV therapeutics have been witnessed recently, with industry investments paving the way for new drugs which target various steps in the HBV life cycle, such as viral entry, replication, and assembly [256]. Although no new therapeutics have reached phase III trials, the development of combination therapies holds promise for developing a finite HBV treatment that aims at achieving a functional cure [257]. Additionally, the emergence of non-governmental groups, such as the World Hepatitis Alliance, and viral hepatitis patients associations, such as ACHIEVE, and national viral hepatitis programs in many countries/regions signifies growing public and political awareness, contributing to the global effort to eliminate hepatitis [18]. 

### 5.5. Strategies for Achieving HBV Elimination by 2030

The development of an effective HBV vaccine has substantially contributed to controlling the spread and progression of HBV-related diseases. However, projections indicate that the persistence of HBsAg prevalence requires intervention. After 2030, numerous countries are expected to make strides toward established goals. Specifically, only three countries are projected to achieve a 90% reduction in HBV incidences, with none reaching a 65% reduction in mortality compared to 2015, and no country is expected to meet all current HBV elimination targets [258].

While notable progress has been made, meeting the WHO’s ambitious goals for HBV elimination by 2030 necessitates continual monitoring and addressing persisting barriers which hinder effective prevention, diagnosis, and treatment interventions. Critical examination of these barriers aims to identify practical solutions, paving the way for a more holistic and equitable approach towards the WHO targets.

Each country and community must develop practical and sustainable strategies tailored to their unique circumstances. The primary focus for the next seven years should be on prevention through universal birth-dose vaccination for all newborns on a global scale. Additionally, health education, tailored to different high-risk populations, is crucial in increasing the awareness of HBV infection risk factors, the consequences of chronic liver disease, and the benefits of screening and effective treatment regimens.

Another critical aspect involves increasing the public awareness of HBV infections through mass screening programs and linking patients to point-of-care facilities [259]. Addressing undiagnosed individuals, known as the “missing millions”, involves combating the barriers of awareness, limited healthcare access, and diagnostic challenges on a global scale. Initiatives, like the World Hepatitis Alliance’s “Find the Missing Millions” campaign, are pivotal in heightening awareness and improving diagnostic endeavors worldwide [260,261].

To ensure the success of elimination strategies, we must make concerted efforts to eradicate the stigma associated with HBV. Due to public misconceptions regarding HBV transmission, individuals with CHB infection often face unjust bias attributed to perceived ‘bad behaviors’, leading to discrimination based on the fear of casual transmission [262].

Moreover, the concept of micro-elimination, widely used in HCV elimination, acknowledges that a universal approach may not suffice to eliminate viral hepatitis globally [263]. Tailored efforts are necessary to address the unique challenges faced by different populations. Micro-elimination initiatives contribute to the overall goal of eliminating viral hepatitis by breaking down the task into manageable segments and addressing the virus in specific contexts.

Various research groups and programs have demonstrated the increased effectiveness and practicality of micro-elimination strategies in achieving the global elimination of HBV [264]. HBV micro-elimination strategies aim to implement measures within populations to prevent transmission, address risk factors, and increase testing, linkage to care, and treatment. These efforts reduce the overall prevalence of HBV and its complications. Micro-elimination programs involve HBV screening, prevention, and identification within specific high-risk groups, such as antenatal screening, infant vaccination, school catch-up vaccination programs, individuals in intermediate-to-high endemic areas, intravenous drug abusers, and prisoners. A study by Chahal et al. demonstrated the cost-effectiveness of HBV micro-elimination strategies like vaccination, prevention, or treatment in reducing the burden of the hepatitis B virus among high-risk, high-prevalence populations [103].

In the context of HBV, imperative initiatives involve implementing programs for the timely administration of HBV vaccines at birth in Africa and other highly endemic regions [241]. Given that a significant number of HBV-infected individuals reside in resource-poor areas, urgent actions are required to enhance access to diagnosis, treatment, and cures. Considering that HBV is a primary contributor to HCC, it is essential to establish screening programs for HCC in individuals infected with HBV [265].

The path to HBV elimination demands sustained commitment, improved coordination among diverse HBV prevention programs, and heightened efforts to diagnose and treat eligible individuals. Given the substantial occurrence of MTCTs, comprehensive testing for HBV in pregnant women is crucial. If infection is detected, promptly administering HBIG to newborns is vital for transmission prevention. Addressing challenges in HBIG administration in Low- and Middle-Income Countries (LMICs), where MTCTs are prevalent, underscores the need for HBV diagnosis and treatment strategies in eligible women of child-bearing age to mitigate MTCTs. In general, the broad-scale diagnosis and treatment of HBV-infected individuals is imperative for achieving HBV elimination [17,239,240].

A pivotal objective for the upcoming decade is the discovery of agents that functionally cure HBV, moving beyond mere suppression.

Countries facing resource constraints require global funding for viral hepatitis initiatives that cover both preventive and therapeutic measures. Furthermore, strengthening programs promoting blood safety and harm reduction, particularly among people who inject drugs, is crucial to curtail HBV transmission. Reducing drug costs, including genetic agents, can elevate treatment coverage for HBV infections. Crucially, the development of curative regimens for HBV will significantly expedite the achievement of the elimination goals [256].

Regular monitoring and evaluation are indispensable for assessing the efficacy of implemented solutions and making necessary adjustments. By addressing these challenges comprehensively, we can expedite progress toward the 2030 goal of eliminating HBV, ultimately saving millions of lives.

## 6. The Road Ahead: Sustaining Progress beyond 2030

To achieve continuous advancement in eliminating HBV beyond 2030, we must establish robust healthcare systems, implement ongoing surveillance, and initiate accessible vaccination initiatives. Integrating these measures into primary healthcare, coupled with fostering community engagement, fortifies prevention efforts [18]. The long-term sustainability of this progress relies on proactively managing HBV, with a particular emphasis on the pivotal role of the HBV vaccine. We must prioritize continuous research endeavors to enhance diagnostics, develop cost-effective treatments, and pursue a functional cure. The prospect of HBV elimination significantly improves if we can develop a ‘cure’, even in a functional form, through a finite course of safe and affordable oral medications, as opposed to the current protracted treatments [266]. Ongoing innovations in drug development technology are expected to expedite progress toward discovering a sterilizing cure for HBV, working in tandem with vaccination programs to facilitate global HBV eradication.

Innovations in vaccine technology, including multi-antigen or therapeutic vaccines, promise to enhance accessibility and overall effectiveness in prevention and treatment. Developing next-generation HBV vaccines with improved attributes, such as thermostability, broader coverage, and longer-lasting immunity, is paramount for eradicating hepatitis beyond 2030. Encouraging innovations in hepatitis vaccine delivery methods becomes crucial.

Advancements in rapid diagnostic tests, point-of-care technologies, and novel antiviral therapies are poised to streamline early diagnosis, simplify treatment procedures, and enhance patient outcomes. The integration of digital tools for data collection and program management optimizes resource allocation and tailors interventions, thereby improving program efficacy. Addressing social determinants of health, reducing health disparities, and fostering global collaboration is indispensable for ensuring long-term success [267].

Despite the efficacy of currently available antiviral therapy in suppressing HBV, patients with advanced hepatic fibrosis or cirrhosis remain susceptible to liver cancer. Monitoring these patients with novel biomarkers for risk stratification is of utmost importance. Future research directions include developing chemo-preventive agents to mitigate hepatocarcinogenesis post-viral controls or cures. While a reduction in advanced liver diseases related to HBV infections is anticipated post-2030, the emergence of other liver diseases, such as metabolic liver disease, alcoholic liver disease, or drug-induced liver injury, is a possibility [268]. Emphasizing health education on risk factors and promoting a healthy lifestyle becomes paramount in the coming decades.

Furthermore, to ensure sustained progress beyond 2030, we must diligently monitor and evaluate initiatives, pinpointing and enhancing specific actions in order to secure achievements. Significant challenges confront the global effort to address viral hepatitis, with sub-Saharan Africa facing obstacles such as limited treatment, minimal testing, and insufficient technical assistance from the WHO, exacerbated by human resource limitations, especially following COVID-19 [18]. Country-specific challenges impede progress, spanning from procuring affordable, high-quality diagnostic tests, antiviral medications, and vaccines to deploying them in remote or underserved areas or populations. Additionally, stringent regulatory frameworks may pose challenges related to approvals, import/export restrictions, and compliance with international standards [264].

These regulations may impede the introduction of new diagnostic tools, treatments, or vaccines, delaying progress in HBV elimination. Even upon achieving the 2030 target, we anticipate substantial ongoing efforts, as not all countries may eliminate viral hepatitis by then, and reaching WHO targets does not ensure universal diagnosis and treatment. The primary goal should be to remove viral hepatitis from the list of major public health threats by 2030 while continuously addressing the issue.

## 7. Conclusions

Approximately 257 million people worldwide are affected by HBV infections, presenting a significant global public health threat, with complications ranking as the seventh-leading cause of mortality. Despite the formidable nature of the task, eliminating HBV by 2030 requires substantial efforts. Ongoing global initiatives encounter feasibility challenges which are influenced by diverse factors, posing specific hurdles in various regions or populations. Implementing HBV vaccinations becomes a cost-effective and pivotal intervention, contributing to an impressive 95% reduction in CHB infections. The global prevalence of HBV has substantially decreased from over 10% in the 1980s to 2.9% in 2020, preserving millions of lives and preventing chronic liver disease and liver cancer cases. Although the global adoption of HBV vaccination is considered successful, addressing existing challenges is crucial for achieving WHO’s ambitious elimination targets, focusing on realizing the vaccine’s full potential. Integral components of the program include prevention strategies, perinatal and neonatal vaccination, post-vaccination monitoring, catch-up vaccinations, and carrier registration. Enhancing the program’s effectiveness involves a continuum of care, increasing public awareness, eradicating stigma and discrimination, integrating with local healthcare services, involving civil society, health policymakers, and funding agencies, and implementing simplified guidelines for diagnosis and treatment. Sustaining progress requires a long-term commitment to sustainability and continuous innovation, with an emphasis on establishing national policies, micro-elimination strategies, and surveillance in order to monitor progress and assess the program’s impact on incidence, mortality, and services.

## Figures and Tables

**Figure 1 vaccines-12-00288-f001:**
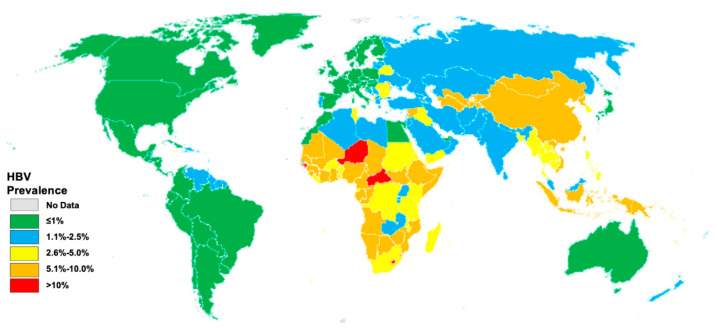
HBV prevalence in all countries in 2022 (all ages) [11]. HBV—hepatitis B virus.

**Figure 2 vaccines-12-00288-f002:**
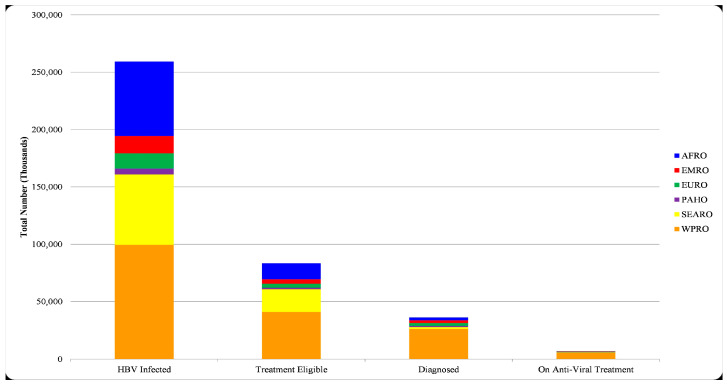
Global and regional HBV cascade of care, 2022 [12]. AFRO–World Health Organization Africa Regional Office; EMRO–World Health Organization Eastern Mediterranean Regional Office; EURO–World Health Organization Europe Regional Office; HBV—hepatitis B virus; PAHO–Pan-American Health Organization; SEARO–World Health Organization South East Asia Regional Office; WPRO–World Health Organization WPRal Office.

**Figure 3 vaccines-12-00288-f003:**
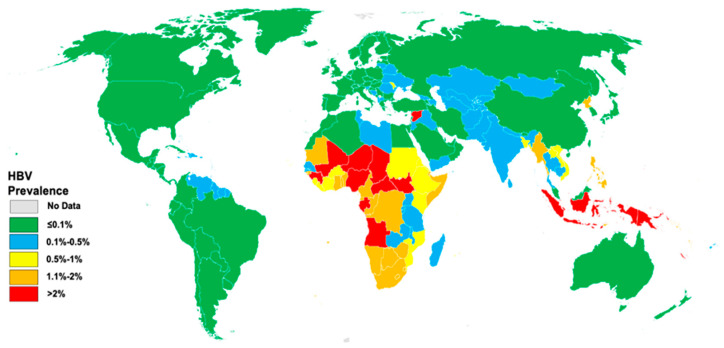
HBV prevalence in all countries (five-year-old children) in 2022 [12]. HBV—hepatitis B virus.

**Table 1 vaccines-12-00288-t001:** WHO 2020 and 2030 targets for eliminating chronic viral hepatitis B and C infection.

Target Area	Baseline 2015	2020 Targets	2030 Targets
Impact targets			
Incidence: New cases of CHB and CHC infections	Between 6 and 10 million infections are reduced to 0.9 million infections by 2030 (95% decline in HBV infections, 80% decline in HCV infections)	30% reduction (equivalent to 1% prevalence of HBsAg among children)	90% reduction (equivalent to 0.1% prevalence of HBsAg among children)
Mortality: HBV and HCV deaths	1.4 million deaths reduced to less than 500,000 by 2030 (65% for both HBV and HCV)	10% reduction	65% reduction
Service coverage targets			
HBV vaccination: childhood vaccine coverage (third dose coverage)	82% in infants	90%	90%
Prevention of HBV MTCTs: HBV birth-dose vaccination coverage or other approach to prevent MTCTs	38%	50%	90%
Blood safety	39 countries do not routinely test all blood donations for transfusion transmissible infections, 89% of donations are screened in a quality-assured manner	All countries have hemovigilance systems in place to identify and quantify viral hepatitis transfusion transmission rates	Reduce rates of transmission by 99% compared with 2020
Safe injections: percentage of injections administered with safety-engineered devices in and out of health facilities	5%	50%	90%
Harm reduction: number of sterile needles and syringes provided per person who injects drugs per year	20	200	300
HBV and HCV diagnosis	<5% of chronic hepatitis infections diagnosed	50%	90%
HBV and HCV treatment	<1% receiving treatment	5 million people receiving HBV treatment, 3 million people received HCV treatment	80% of eligible persons with CHB infection treated, 80% of eligible persons with CHC infection treated

The abbreviation: HBsAg: hepatitis B virus surface antigen; HBV: Hepatitis B virus; HCV: hepatitis C virus; CHB: chronic hepatitis B; CHC: chronic hepatitis C.

**Table 2 vaccines-12-00288-t002:** Four strategies to universal hepatitis B vaccinations in infants.

Maternal Screening	Infants Receive	Efficacy	Cost	Example	
	Vaccine	HBIG			
Yes (HBsAg and then HBeAg)	Yes	HBeAg-positive mothers’ infants only	Higher	Higher	Taiwan
Yes (HBsAg only)	Yes	All HBsAg-positive mothers’ infants	Highest	Highest	US
Yes (HBeAg only)	Yes	HBeAg-positive mothers’ infants only (2 doses)	High	Highest	Japan
No	Yes	No	Modest	Low	Thailand

Abbreviations: HBsAg: hepatitis B surface antigen, HBeAg: hepatitis B e antigen, HBIG: hepatitis B immune globulin.

## Data Availability

Not applicable.

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
