# Peer review of "Global Perspectives on the Hepatitis B Vaccination: Challenges, Achievements, and the Road to Elimination by 2030"

_vaccines, 2024, doi:10.3390/vaccines12030288_

Round 1

Reviewer 1 Report

Comments and Suggestions for Authors

This is a good manuscript that give a detailed and comprehensive overview of the achievements and challenges ahead for Hepatitis B Vaccination. The review is well done and will most probably become a very useful resource for researchers in this field in years to come. The WHO objectives are clearly laid out and the authors have done a good job of collecting evidence, and interstesting case studies from across the globe. The review includes a thourough bibligaphy that to the best of my knowlegdge is appropriate. The manuscript is very long but I leave the decision whether to shorten it to the editors of the journal. 

Before this review paper will be published I really want the authors to take into account the following points, in order to futher enhance the quality of the final manuscript. 

1) Please give greatere details in the section of Methodology (lines  164 to 178) How many documents did you end up etc. This is really necessary.

2) There are some interesting policy suggestions scattered throughout the manuscript, in particular in section 5 (see for example in lines 848 onwards, 981-984 and 993-936 amongst others). I would like the authors to summarise the policy suggestions into a couple of paragraphs that should go into the section Conclusions. 

3) Please clarify better the caption that is supposed to explain Figure 2, it is really unclear as it is and should be redone. I had problems understanding what exactly the authors wanted to communicate. 

4) Table 1, please put in the caption what MTCT stands for (mother to child transmission), to remind the reader

5) Line 970 what does Post-2050 refer to in this context? Please explain

Comments on the Quality of English Language

Overall the manuscript is clearly written but there are odd little things here and there. 

Author Response

Dear Reviewer

I sincerely appreciate your thorough review of our manuscript titled "[Global Perspectives on Hepatitis B Vaccination: Challenges, Achievements, and the Road to Elimination by 2030]" with Manuscript ID: [vaccines-2886225]. Your insightful comments have been invaluable in enhancing the quality of our work. I would like to express my gratitude for your positive assessment and constructive suggestions.

Here is a summary of the actions taken in response to your recommendations:

  1. Methodology Section:

   - Action: The methodology section has been revised with additional details, addressing your concern regarding the number of documents reviewed. We believe this enhancement provides a clearer understanding of our research process.

  1. Policy Suggestions in Section 5:

   - Action: The scattered policy suggestions in Section 5 have been consolidated into a concise summary within the Conclusions section. This modification aims to improve the overall coherence and impact of our manuscript.

  1. Clarification of Figure 2 Caption:

   - Action: While the abbreviations in Figure 2 are consistent with WHO standards, we have added more detailed explanations in the text to improve clarity. We appreciate your suggestion and believe this enhances the reader's comprehension.

  1. Table 1 Caption (MTCT):

   - Action: The Mother-to-Child Transmission inside the table is left without an abbreviation/caption. The abbreviation was later added to the text.

  1. Line 970 (Post-2050):

   - Action: We acknowledge the typo error, and Line 970 (now line 1085) has been corrected to "Post-2030" to accurately reflect the intended context. Thank you for pointing out this oversight.

Your insightful feedback has played a pivotal role in refining our manuscript, and we are grateful for your time and expertise. We believe these revisions significantly contribute to the clarity and overall quality of the paper.

Thank you once again for your valuable comments and commitment to improving our work.

Best regards,

Said Al-Busafi 

Reviewer 2 Report

Comments and Suggestions for Authors

The paper provides a comprehensive review of the global status of Hepatitis B (HBV) vaccination efforts, emphasizing the critical role of vaccines in combating HBV infection and the ambitious goal of eliminating HBV as a public health threat by 2030, as set by the World Health Organization (WHO). It thoroughly examines the disparities in vaccination coverage worldwide, the significant impact of vaccinations on reducing HBV prevalence, and the formidable challenges that must be addressed to meet WHO's targets. Additionally, the paper discusses the importance of addressing vaccine hesitancy, improving access to vaccination in underprivileged areas, and the need for global cooperation to overcome logistical and financial barriers to vaccine distribution.

Drawbacks:

1. While the paper covers the evolution and effectiveness of HBV vaccines, it could benefit from a deeper exploration of emerging vaccine technologies and their potential to enhance vaccination strategies.

2. The paper might have an imbalance in geographical coverage, with a need for more detailed case studies from regions with low vaccination rates to understand unique local challenges.

3. A more detailed economic analysis of vaccination programs could enrich the discussion, especially in low-income countries where financial constraints are a significant barrier to vaccine implementation.

Recommendations:

1. Update the review to include recent advancements in vaccine technology and delivery systems that could improve immunization rates and outcomes.

2. Include in-depth analyses of regions with successful and struggling vaccination campaigns to identify best practices and lessons learned.

3. To support policy-making and funding allocation decisions, add a comprehensive economic evaluation of HBV vaccination programs, including cost-effectiveness studies.

4. Emphasize the importance of strengthening partnerships between governments, international organizations, NGOs, and the private sector to enhance vaccine access and affordability.

5. Highlight the need for innovative public awareness campaigns to combat vaccine hesitancy and misinformation, especially in regions with low vaccination rates.

Author Response

Dear Reviewer

I extend my sincere gratitude for your insightful comments and recommendations on our manuscript titled "[Global Perspectives on Hepatitis B Vaccination: Challenges, Achievements, and the Road to Elimination by 2030] and Manuscript ID (vaccines-2886225)." Your constructive feedback has significantly contributed to enhancing the overall quality of our work. Your detailed insights were instrumental in refining the paper, and we appreciate the time and effort you dedicated to the review.

Here are the actions taken in response to your valuable comments:

  1. Deeper Exploration of Emerging Vaccine Technologies:

   - Action: Section 3.6, focusing on recent advances in HBV vaccination (lines 406 to 480), has been rewritten with a more comprehensive exploration of emerging vaccine technologies. This revision aims to provide a deeper understanding of evolving vaccination strategies.

  1. Geographical Coverage and Case Studies:

   - Action: Throughout the manuscript, including Section 3.7.2 (HBV mass vaccination) and Section 4 (Global Impact of HBV Vaccination Programs), we have incorporated in-depth analyses of regions with both successful and struggling vaccination campaigns. This includes identification of best practices and lessons learned.

  1. Comprehensive Economic Evaluation:

   - Action: A new section (Section 3.5) has been added, presenting a comprehensive economic evaluation of HBV vaccination programs. This includes detailed cost-effectiveness studies, especially focusing on low-income countries where financial constraints pose significant barriers.

  1. Emphasis on Strengthening Partnerships:

   - Action: Emphasis on the importance of strengthening partnerships between governments, international organizations, NGOs, and the private sector to enhance vaccine access and affordability has been added. This information is now highlighted in various sections, including Section 5.2 (line 947), Section 5.4 (lines 1038 to 1040), and the conclusion.

  1. Innovative Public Awareness Campaigns:

   - Action: Acknowledging the importance of improving public awareness, the manuscript has alluded to the role of healthcare workers and emphasized public awareness in different sections, such as Section 4 (lines 823 to 825), Section 5.5, and the conclusion (lines 1101 to 1106).

Your thoughtful recommendations have significantly enriched our manuscript, and we are grateful for your valuable contribution.

Thank you once again for your time, expertise, and commitment to ensuring the excellence of our work.

Best regards,

Said Al-Busafi